# Stray Dogs and Public Health: Population Estimation in Punjab, India

**DOI:** 10.3390/vetsci9020075

**Published:** 2022-02-10

**Authors:** Gurlal S. Gill, Balbir B. Singh, Navneet K. Dhand, Rabinder S. Aulakh, Michael P. Ward, Victoria J. Brookes

**Affiliations:** 1Centre for One Health, Guru Angad Dev Veterinary and Animal Sciences University (GADVASU), Ludhiana 141001, India; gillkangar@pau.edu (G.S.G.); rsaulakh@gadvasu.in (R.S.A.); 2Krishi Vigyan Kendra, Faridkot, 151203, Punjab Agricultural University, Ludhiana 141004, India; 3Sydney School of Veterinary Science, The University of Sydney, Camperdown, NSW 2006, Australia; navneet.dhand@sydney.edu.au (N.K.D.); michael.ward@sydney.edu.au (M.P.W.); victoria.brookes@sydney.edu.au (V.J.B.)

**Keywords:** India, Punjab, roaming-dog, stray dog count, public health

## Abstract

The overpopulation of stray dogs is a serious public health and animal welfare concern in India. Neglected zoonotic diseases such as rabies and echinococcosis are transmitted at the stray–dog human interface, particularly in low to middle-income countries. The current study was designed to estimate the stray dog populations in Punjab to enhance the implementation of animal birth and disease (for example, rabies vaccination) control programs. This is the first systematic estimation of the stray dog population using a recommended method (mark–re-sight) in Punjab, India. The study was conducted from August 2016 to November 2017 in selected villages or wards in Punjab. For the rural areas, 22 sub-districts in each district were randomly selected, then one village from each of the 22 selected sub-districts was selected (by convenience sampling). For urban areas, 3 towns (less than 100,000 human population) and 2 large cities (more than or equal to 100,000 human population) were randomly selected, followed by convenience selection of two wards from each of the 5 selected towns/cities. To estimate the dog population size, we used a modified mark–re-sight procedure and analysed counts using two methods; the Lincoln–Petersen formula with Chapman’s correction, and an application of Good–Turing theory (SuperDuplicates method; estimated per km^2^ and per 1000 adult humans and were compared between localities (villages vs. towns), dog sex (male vs. female) and age group (young vs. adult) using linear mixed models with district as a random effect. The predicted mean (95% CI) count of the dogs per village or ward were extrapolated to estimate the number of stray dogs in Punjab based on (a) the number of villages and wards in the state; (b) the adult human population of the state and (c) the built-up area of the state. Median stray dog populations per village and per ward using the Lincoln–Petersen formula with Chapman’s correction were estimated to be 33 and 65 dogs, respectively. Higher estimates of 61 per village and 112 per ward are reported using the SuperDuplicates method. The number of males was significantly higher than the number of females and the number of adult dogs was about three times the number of young dogs. Based on different methods, estimates of the mean stray dog population in the state of Punjab ranged from 519,000 to 1,569,000. The current study revealed that there are a substantial number of stray dogs and a high number reside in rural (versus urban) areas in Punjab. The estimated stray dog numbers pose a potential public health hazard in Punjab. This impact requires assessment. The estimated stray dog numbers will help develop a dog population and rabies control program in which information about the logistics required as well as costs of implementing such programmes in Punjab can be incorporated.

## 1. Introduction

Dogs are the first species to have been domesticated and share a close cultural, social and economic association with humans [1]. Pets are treated as family members in the Western world and approximately 50% of households keep pets [2]. Domestication of dogs has been shown to provide many benefits to humans [3]. For example, pet ownership is associated with a decreased prevalence of depressive symptoms [4]. It has been reported that pet owners visit their doctor less often, use fewer medications and have lower blood pressure and cholesterol levels than non-pet owners [5]. Dogs have roles in the management of many psychological, psychiatric and biomedical conditions in humans [6,7]. However, dog domestication—in particular, the overpopulation of stray dogs—can have negative impacts on public health and animal welfare.

A stray dog is defined as any dog in a public area that is not under direct human control. Therefore, this term encompasses unowned and community-owned dogs but excludes dogs on leashes or under direct human control at the time of a survey (WSPA, 2009). Stray dogs can be either previously owned dogs [8,9] or can be feral dogs (wild dogs) that have never been owned [10]. Factors such as easy availability of food, lack of predators, low number of competitors, and the ease with which a breeding partner can be found can lead to rapid increases in stray dog populations [11].

In contrast to the health benefits provided by pet dogs, stray dogs contribute to environmental pollution, dog bite incidence and can act as reservoirs of many important zoonotic parasites (for example, *Toxocara*, *Ancylostoma* and *Echinococcus*) via faecal contamination of soil and water [12,13,14], and infectious diseases (for example, rabies, salmonellosis). The faecal shedding of pathogens by stray dogs contaminates the environment [15], which is of substantial public and animal health concern. Stray dogs also contribute to incidents such as bites and accidents and damage wildlife populations [16,17,18,19]. In addition, stray dogs have a substantial negative economic impact at tourist destinations [20].

The overpopulation of stray dogs is an important animal welfare and public health concern in India [21,22]. It has been reported that 92–97% of human rabies deaths occur in India following bites from infected dogs, of which 60% have been reported from stray dogs [23,24]. Stray dogs transmit many important zoonotic pathogens in India [25]. Large numbers of dogs have been reported to roam the streets in many parts of the country [26,27,28]. These high numbers of stray dogs are due to large amounts of edible waste available on the streets, cultural tolerance of stray dogs and a lack of consistently employed sustained birth control programs [29].

Mass culling of dog populations was historically used to control rabies in India [30], but this has been replaced by animal birth control (ABC) programs under the Prevention of Cruelty to Animals Act, 1960 and the Animal Birth Control Rules, 2001 (http://envfor.nic.in/legis/awbi/awbi13.pdf, accessed on 15 March 2019). Within an ABC program, street dogs are sterilized, vaccinated for rabies and subsequently released into the same area from which they were captured. The ABC program has been implemented in small areas in India with effective results [26,30,31,32]. For example, a decline in human dog-bite cases and stray dog numbers in Jaipur, India, during and after the implementation of the ABC program [27,31] has been reported.

According to official data, there are 17.14 million stray dogs in India [33]. This estimate is based on the collection and compilation of information from veterinary personnel based on their areas of work, and passive surveillance. Reliable and repeatable scientific methods have not been used to derive this estimate, and so it might be inaccurate and likely lacks precision.

Globally, recommended methods for the estimation of the size of stray dog populations include total or direct counts, and mark–resight or capture–recapture methods [34,35]. However, total or direct counts are not practical over large geographical areas and dog populations [34,35]. In the past, several capture–recapture enumeration methods have been used to drive free-roaming dog population estimates [36]. Out of these, Lincoln–Petersen’s formula with Chapman’s correction have been commonly used to drive these estimates. Recent studies indicate the usefulness of the SuperDuplicates method to estimate dog counts, particularly for two sample surveys conducted on consecutive days [36].

Stray dog population estimates using the recommended methods have been carried out in countries neighbouring India such as Bangladesh [37,38] and Bhutan [38], as well as in other parts of the country such as Rajasthan [27] and Maharashtra [28], but not in the Punjab state of India. Additionally, stray dog age- and gender-specific data necessary to facilitate ABC programs are not available from the official estimates. Therefore, the current study was planned to estimate the stray dog population size in Punjab, India. Demographic data—as well as some health indices (lameness, skin conditions, and open wounds)—were also recorded.

## 2. Methods

### 2.1. Study Area

The study was conducted in Punjab State in the northwest region of India (latitudes 29.30° North to 32.32° North and longitudes 73.55° East to 76.50° East) between August 2016 and November 2017 (16 months). Punjab has a land area of 50,362 km^2^ comprising 48,265 km^2^ rural and 2097 km^2^ urban areas, including 237 towns and cities (http://punjab.gov.in/know-punjab, accessed 15 March 2019). Punjab has a human population of 27,743,338 with a density of 551 people/km^2^.

### 2.2. Selection of Villages and Wards

The state has 22 districts divided into 81 tehsils (sub-districts), containing 12,581 villages and 217 towns and cities. Of the towns and cities, 18 have >100,000 humans and 199 have <100,000 humans (http://www.citypopulation.de/php/india-punjab.php, accessed on 15 March 2019). For village selection from the rural areas, 22 sub-districts (one from each district) were randomly selected, followed by convenience selection of one village from each of the selected sub-districts (i.e., a total of 22 villages). For municipal ward selection from the urban areas, 3 towns (with <100,000 humans) and 2 large cities (with >100,000 humans) were randomly selected followed by convenience selection of two wards each from the selected towns and cities (a total of 10 wards). The official adult human population data were collected from the respective village heads or municipal councillors (Appendix A). The locations of villages and wards are shown in Figure 1.

### 2.3. Stray Dog Count

The current study was conducted between August 2016 and November 2017. Stray and pet dogs were differentiated according to WSPA guidelines [39]. The purpose of this study was to estimate the size of the stray dog population to inform ABC programs. Therefore, an announcement was made before the stray dog count in the relevant villages and wards about the stray dog counting to encourage people to leash their pet dogs, to avoid over-estimation of the population.

We used a mark–re-sight procedure [40] with modifications [37] to estimate the stray dog populations. In brief, a two–three-person survey team was formed and formally trained to count dogs in all the selected villages and wards. On Day 1, between 5.30–6.30 a.m. in the summer and 6.30–7.30 a.m. in the winter, the team marked stray dogs in the street observed during walks using sprayed water-soluble red paint. Dogs were not physically restrained or captured. In addition, data related to demography, health status (gross examination without restraining for the presence of open wounds, skin disease and skeletal deformity), sex and age (young or adult by observing the external genitalia) were also recorded. The entire village/ward was covered using a pre-determined route which was recorded during the stray dog estimation using Arc GIS explorer10.2.1 software. The area of village/ward covered during the stray dog estimation was also recorded using Arc GIS expo 10.2.1 software. For ease of marking and to make it dog-friendly, we used baits (dog biscuits, dog feed) to feed stray dogs.

Dogs were photographed if they could not be marked (for example, if they ran away) or had a reddish, black or mixed colour coat. All village or ward areas, including garbage dumps, were systematically searched. The number of dogs marked with colour or which were photographed on Day 1 was recorded and represented ‘n1’ within the mark–re-sight framework.

The Day 1 procedure was repeated on Day 2 in all selected villages and wards. The same start times and routes were followed. The marked dogs encountered on Day 2 were recorded separately. To ensure identification and to avoid double-counting, additional matching with photographs was conducted if required.

### 2.4. Population Size Estimates

#### 2.4.1. Lincoln–Petersen’s Formula with Chapman’s Correction

The total number of dogs counted on Day 2 was designated ‘n2’ and the number of re-sighted (marked or photographically identified) dogs was designated ‘m’ in the mark–re-sight framework. The sizes of the stray dog populations were estimated using the Lincoln–Petersen formula with Chapman’s correction [41] according to Equation (1) in which
(1)$$N=[\frac{\left(n1+1\right)\left(n2+1\right)}{m+1}-1]$$
(2)$$var\left(N\right)=\left[\frac{\left(n1+1\right)\left(n2+1\right)\left(n1-m\right)\left(n2-m\right)}{{\left(m+1\right)}^{2}\left(m+2\right)}\right]$$
(3)$$95\%\;confidence\;interval\;\left(CI\right)=N\pm 1.965\sqrt{var\left(N\right)}$$

*N* is the estimate of the total population size, *n*1 is the total number of dogs marked on Day 1, *n*2 is the total number of dogs sighted on Day 2, and *m* is the number of marked dogs re-sighted on Day 2. An approximate unbiased variance of *N* was estimated by using Seber’s formula [40] (Equation (2)). The 95% confidence interval for *N* was estimated according to Equation (3).

#### 2.4.2. Recapture Probability

The recapture probability or the detectability rate (*r*) was estimated according to Equation (4) in which the number of dogs that were re-sighted on the second day (*n*2) was divided by the estimated population (*N*).
(4)$$r=\frac{n2}{N}$$

#### 2.4.3. SuperDuplicates Method

The dog counts were also estimated using this method. Population size estimates are based on the presence or absence of each observed dog in repeated samples. The dog population size was estimated using the online tool https://chao.shinyapps.io/SuperDuplicates/ [42] (accessed on 28 October 2021). Data included the total number of observed dogs (‘observed species’ in this framework), the number of dogs that were sighted only once (‘uniques’), and the number of days (‘samples’).

### 2.5. Statistical Analyses

All statistical analyses were conducted using R statistical program (R statistical package version 3.4.3, R Development Core Team, http://www.r-project.org; accessed on 28 October 2021).

Descriptive statistics and frequency distributions of gross health examinations, sex and age of the stray dogs were generated for villages and wards. The stray dog population for each village and ward/km^2^ was calculated by dividing the stray dog count in the village or ward by the area of the village or ward in km^2^. Similarly, stray dog population/1000 adult human population (based on the latest voter lists) for each village and ward was estimated by dividing the stray dog count in each village and ward by their respective adult human population.

Stray dog populations (both per km^2^ and per 1000 adult humans) were compared between villages and wards (urban areas in towns and cities) using linear mixed models with the variable location (village or ward) as a fixed effect and district as a random effect. Similar analyses were conducted using the sex and age groups as fixed effects. Model assumptions were evaluated using residual diagnostics and the outcome variables were log-transformed (if required) to meet the assumptions. Predicted group means and differences between group means were back-transformed for presentation. Note that the predicted means calculated on the log scale become geometric means on back-transformation whereas the differences between the group means become a ratio after back-transformation.

The mean (95% CI) stray dog population at the state level was estimated separately for rural and urban areas by multiplying (a) the predicted mean (95% CI) count of the dogs per village or ward with the total number of villages and wards in the state, respectively; (b) the predicted mean (95% CI) dog count per 1000 adult human population with the total adult human population of the state; and (c) the predicted mean (95% CI) dog count per km^2^ residential built-up land area with the total built-up area of the state [43]. We included industrial areas with the urban built-up land area for estimating the urban built-up area. Therefore, a separate analysis was also conducted after excluding the industrial area from the urban built-up area to estimate the number of stray dogs residing in the urban areas.

## 3. Results

### 3.1. Number and Density of Stray Dogs in Village(s)/Ward(s)

Detailed information on the number of stray dogs estimated using Lincoln–Peterson’s formula with Chapman’s correction in the selected villages and wards is shown in Appendix A. Overall, we recorded 1011 (614 rural and 397 urban) stray dogs on day 1 (*n*1), 1002 (606 rural and 396 urban) stray dogs on day 2 (*n*2), and 664 (440 rural and 224 urban) stray dogs were re-sighted (*m*). The overall detectability rate (*r*) or recapture probability of stray dogs was 65.7% (71.7% in rural areas compared to 56.5% in urban areas).

Detailed information on the number of stray dogs estimated in the selected villages and wards using the SuperDuplicates method is shown in Appendix A. The incidence data indicated that there are 2076 observed dogs (‘observed species’ in this framework) and 1349 dogs that were sighted only once (‘uniques’).

Summary statistics of dog populations per village and ward, km^2^ and 1000 adult humans estimated using both the methods are presented in Table 1. Dog population estimates were not significantly different between rural and urban areas using Lincoln–Petersen’s formula with Chapman’s correction or SuperDuplicates method (Table 2). The population of male dogs was significantly higher than that of female dogs and the population of adult dogs was higher than that of young dogs in both Lincoln–Petersen’s formula with Chapman’s correction and SuperDuplicates estimates (Table 2).

### 3.2. Stray Dog Count in Punjab

Detailed information on the number of stray dogs in Punjab estimated from numbers derived using the Lincoln–Peterson formula with Chapman’s correction is presented in Table 3. The mean dog population in the state was estimated to be 519,000 (/1000 human population)—868,000 (/km^2^). The population of rural dogs was estimated to be 329,000 (/1000 human population)—576,000 (/km^2^) and was higher than the estimated population of urban dogs (190,000–292,000). A mean stray dog count of 276,321 (95% CI: 199,685–354,036) was estimated for the urban areas after excluding the industrial area from the urban built-up area. We estimated a human to stray dog ratio of 38:1 and 42:1 in rural and urban areas, respectively.

Detailed information on the number of stray dogs in Punjab is using the SuperDuplicates method is also presented in Table 3. The mean dog population in the state was estimated to be 944,000 (/1000 human population)—1,569,000 (/km^2^). The population of rural dogs was estimated to be 622,000 (/1000 human population)—1,071,000 (/km^2^) and was higher than the estimated population of urban dogs (265,000–498,000). A mean stray dog count of 471,689 (95% CI: 330,290–613,087) was estimated for the urban areas after excluding the industrial area from the urban built-up area. We estimated a human to stray dog ratio of 28:1 and 32:1 in rural and urban areas, respectively.

### 3.3. Abnormalities Detected in Stray Dogs

Abnormalities detected in stray dogs are presented in Table 4. During the stray dog count, we estimated gross health abnormalities in 4.4% (38/845) and 4.7% (33/701) of the dogs residing in rural and urban areas, respectively (chi square = 0.35, *p* = 0.85). Gross health abnormalities such as wound (chi square = 0.12, *p* = 0.72), skin diseases (chi square = 0.08, *p* = 0.77) and emaciation (chi square = 0.007, *p* = 0.95) did not differ significantly between the rural and urban areas.

## 4. Discussion

As far as we are aware, this is the first systematic estimation of the stray dog population in Punjab. According to 2012 official data, there were estimated to be 305,482 stray dogs in Punjab [44]. However, we estimated the stray population to be 519,000 to 1,569,000 using mark–re-sight and the Superduplicates method, which is much higher than the official estimates. Mark–re-sight is considered a practical way to accurately estimate the number and distribution of a stray dog population if the assumption of a closed population is fulfilled [40,45]. In the present investigation, marking and subsequent counting events were completed within two days and thus the assumption of a closed population was likely valid because the period between the counting events was very short. Recent studies indicate that the SuperDuplicates method is also useful to estimate dog counts, particularly for two sample surveys conducted on consecutive days [36]. The large difference between our estimate and the official estimate could be explained by the animal husbandry department having estimated the stray dog population size in 2011; there might have been a substantial increase in the stray dog population since then due to the lack of effective stray dog control programs. Differences in the methods used may also be responsible for differences in stray dog estimations. This is consistent with the findings of researchers in other areas [46].

Our estimate of a higher male to female dog population ratio is similar to the findings of other national and international studies [27,37,47,48]. Whilst this might be due to methodology differences or measurement error, many factors that influence higher male survivability or higher mortality in female dogs favour a high population of male dogs. Populations with unequal sex ratios have a lower reproduction potential compared to equal sex ratio populations [49]. The sex ratio of the owned dog population might be responsible for this phenomenon. It has been reported that owners and farming communities prefer male dogs due to guarding requirements, reduced nuisance behaviour during oestrus, and avoiding unwanted puppies when compared to adult females [50,51]. However, this needs to be further investigated.

The higher adult population estimated in the current study is consistent with many other international dog count estimates [27,48]. High early life mortality has been reported among free-ranging puppies from India [52], but as far as we are aware, the life expectancy of stray dogs in India is unknown. Whilst the average life expectancy among companion dogs can be relatively long—for example, it has been reported to be 13.7 years in Japan [53], we expect it to be much shorter in free-roaming dogs. Further studies need to be conducted to understand adult-juvenile dog ratios in the state.

We extrapolated the dog population to the state level based on the number of villages and wards, residential (built-up) areas and the adult human population residing in the surveyed areas. The dog population estimates were highest based on the built-up area and were lowest based on the adult human population. It needs to be ascertained which method provides a better estimate because a “gold standard” was not available in the current study. We are not aware of any previous studies using these methods to extrapolate dog population data. Further investigations are required to evaluate the accuracy of these methods. The estimates using a mark–re-sight method to estimate the dog population size observed a re-sight probability of stray dogs of 66% (rural 72% and urban 56%). Similar re-sight (recapture) probabilities between 61% and 63% have been estimated in some previous studies [37,54], whereas a lower probability of 46–49% has also been reported [38].

We estimated a human to stray dog ratio of 28–38:1 and 32–42:1 in rural and urban areas, respectively. This is higher than the World Health Organizations average estimate of 10 people/dog. Human to stray dog ratios have been reported to be 15:1 in Bhutan [38], 4.7:1 in Kathmandu, Nepal [55], 5.2:1 in Shimotsui, Japan, 23:1 in Timor Leste [54] and 828:1 and 120:1 in Bangladesh [37,50]. The human–stray dog ratio estimated in the current study is lower than those reported in Bangladesh but higher than Bhutan, Nepal and Japan, which might be due to socio-cultural and human population density differences in different countries. For example, the Bangladeshi study was conducted in Dhaka [37], one of the most densely populated cities in the world.

We estimated the dog density in rural and urban areas to be 310–577 and 256–437/km^2^ built-up area, respectively. A range of dog densities have been reported in previous studies: 185 free-ranging dogs/km^2^ from West Bengal, India [47], 57 free-ranging dogs/km^2^ in Mumbai, India [28], 225 stray dogs/km^2^ in Shimotsui, Japan [55], 2930 stray dogs/km^2^ in Kathmandu, Nepal [55], 52 dogs/km^2^ [37] and 14 dogs/km^2^ [50] in Bangladesh. These densities have been estimated from small areas, and therefore, might not be representative of the country or state level. Practices such as waste disposal that affect the availability of food, as well as space, can influence the stray dog populations within these areas.

This study had several limitations. We only surveyed 22 villages and 10 wards of Punjab. Stray dog estimation from additional villages and wards could have improved the precision of our estimates. The target population was stray dogs to inform dog control programs. We faced a problem of dogs running away before being marked. Our solution was to use food baits to facilitate marking Day 1 sighted dogs. In a vaccination program in Western India, the use of a food bait enabled access to 80% of the sighted dogs in a rabies vaccination program [56]. Further, we believed that food baits were essential to avoid dog bites to researchers. However, the effect of food baits on the overall dog count needs to be considered. Both positive (food baits) and negative (marking) reinforcements could have influenced dog behaviour and the probability of being re-sighted on Day 2, affecting the recapture counts. If the baiting on Day 1 artificially inflated the number of dogs re-sighted on Day 2 (because dogs returned for food), this would have resulted in an underestimate of dog numbers. If the opposite happened and these dogs actively avoided the counters (for fear of being marked again) on Day 2, then there would be an overestimation of dog numbers. We believe that the combination of these two potential errors is likely to have resulted in no substantial bias. The influence of physical marking of street dogs during counts has not been previously considered [57]. An alternative is to use photographs only, for example, using an iPad or tablet [58]. The mean dog counts per village/ward were higher using the SuperDuplicates method compared to Lincoln–Petersen’s formula with Chapman’s correction. Further, we only used two surveys to achieve this estimate. We recommend studies in the future to refine these estimates.

Although we took the utmost care in estimating only the stray dog population, some proportion of free-roaming, owned dogs could have been counted as stray dogs. The stray dogs were not captured and conditions such as tick infestation and lactation status could not be estimated. Moreover, dogs were not physically restrained or captured for gross health examinations. Therefore, the gross-health abnormalities are likely to be under-reported. The number of lactating females might serve as an indicator of the population turnover of stray dogs. We purposively selected villages and wards and believe that random selection at the village and ward level is unnecessary because villages and wards in Punjab are very homogenous with respect to socio-economic indicators and other factors such as human population density. However, the presence of waste dumps, natural carcass disposal sites and restaurants in some of the selected areas might have biased the results. In addition, although we took care to count re-sighted dogs through marking and photographic records, a small mismatch error was possible.

We used a recommended mark–re-sight approach and GPS-based measurements of residential areas to improve the accuracy of the dog population estimates in Punjab, India.

Overall, we estimated that there are 519 to 1569 thousand stray dogs with a higher number residing in rural areas compared to urban areas, more male than female dogs, and more adult than young stray dogs in Punjab. We also estimated the burden of a range of health abnormalities in this population.

## 5. Conclusions

The estimated stray dog numbers pose a potential public health hazard in Punjab. Our estimates are higher than previous estimates and therefore, the potential impact and the impact of stray-dog population control requires assessment. The estimated stray dog numbers can be used to guide the development of a dog population control program in Punjab, especially if detailed information about resources (including costs) and logistics required can be included.

## Figures and Tables

**Figure 1 vetsci-09-00075-f001:**
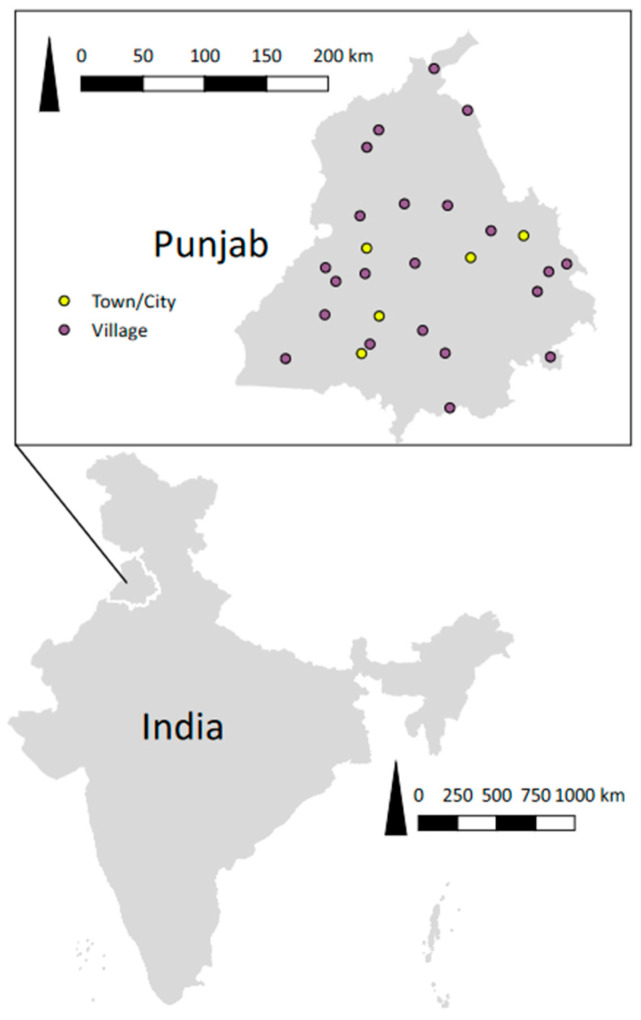
Study area showing 22 villages and 5 cities/towns surveyed in the Punjab state of India to estimate dog populations between August 2016 and November 2017. QGIS 3.6.0 was used to create the figure (qgis.org; accessed on 27 June 2019). The shapefiles are publicly available from diva-gis.org (accessed on 27 June 2019; India and Punjab) or created by the authors (village and cities/towns).

**Table 1 vetsci-09-00075-t001:** Summary statistics of the number of stray dogs in 22 rural and 10 urban areas of the Punjab state of India surveyed between August 2016 and November 2017.

Count	Category	Min	Q1	Median	Q3	Max	Mean	Min	Q1	Median	Q3	Max	Mean
		Lincoln–Petersen Formula with Chapman’s Correction						Super Duplicates Method					
Dogs/village or ward	Rural	13	26	33	50	71	38	24	49	61	97	124	71
	Urban	21	40	65	78	151	68	35	65	112	147	254	116
Dogs/km^2^	Rural	150	238	291	371	506	310	282	438	558	698	952	577
	Urban	59	159	244	340	500	256	101	258	405	647	784	437
Dogs/1000 adult humans	Rural	13	22	26	35	43	27	25	41	49	60	79	50
	Urban	9	17	20	37	66	28	16	29	33	60	142	50

**Table 2 vetsci-09-00075-t002:** Comparison of stray dog populations (/km^2^ and /1000 adult human population) between locality, sex and age groups using linear mixed models based on a survey conducted in Punjab, India between August 2016 and November 2017.

Count	Category	Mean Population	95% CI	Ratio	*p*-Value	Mean Population	95% CI	Ratio	*p*-Value
		Lincoln–Petersen Formula with Chapman’s Correction				Super Duplicates Method			
Dogs/km^2^	Rural	310	262, 358	1.21	0.23	577	489, 666	1.32	0.08
	Urban	256	185, 328			437	306, 568		
Dogs/1000 adult human population	Rural	27	22, 31	1.04	0.66	51	40, 59	1.16	0.34
	Urban	26	18, 32			44	31, 55		
Dogs/km^2^	Male	165	142, 188	1.24	0.03	305	261, 349	1.21	0.03
	Female	133	110, 156			252	208, 296		
Dogs/1000 adult humans	Male	15	12, 17	1.25	0.01	27	22, 31	1.23	0.04
	Female	12	10, 13			22	18, 25		
Dogs/km^2^	Young	69	45, 93	0.31	<0.001	133	86, 180	0.32	<0.001
	Adult	224	200, 248			418	371, 465		
Dogs/1000 adult humans	Young	6	5, 7	0.29	0.001	12	9, 13	0.32	<0.001
	Adult	21	17, 24			38	31, 43		

**Table 3 vetsci-09-00075-t003:** Estimated number of stray dogs (in thousands) in Punjab state of India, extrapolated from a study conducted in 22 villages and 10 wards of the state in 2016–2017.

Category	State-Level Data	Total Number of Dogs (1000 s)					
		Lincoln–Petersen Formula with Chapman’s Correction			Super Duplicates Method		
		Mean	LCL	UCL	Mean	LCL	UCL
	Number of villages and wards						
Rural	12,581	465	351	551	868	642	1032
Urban	2496	157	100	206	265	170	205
Total	12,581 and 2496	622	451	757	1133	812	1237
	Residential built up area (km^2^)						
Rural	1857	576	487	665	1071	908	1237
Urban	1139	292	211	374	498	349	647
Total	2997	868	698	1039	1569	1257	1884
	Adult human population						
Rural	12,201,170	329	268	378	622	488	720
Urban	7,315,518	190	132	234	322	227	402
Total	19,516,688	519	400	612	944	715	1122

**Table 4 vetsci-09-00075-t004:** Frequency table for abnormalities recorded in stray dogs residing in rural (*n* = 845) and urban (*n* = 701) areas in Punjab during a survey conducted 2016–2017.

Type of Abnormality	Frequency	Relative Frequency (%)
Rural areas		
Wounds	7	0.82
Skin diseases (scabies/mange/inflammation)	11	1.30
Fractures	2	0.23
Hind-limb paralysis	3	0.35
Emaciation	14	1.65
Mandibular deformity	1	0.11
Gross health abnormalities	38	4.40
Urban areas		
Wounds	7	0.99
Skin diseases (scabies/mange/inflammation)	8	1.14
Fractures	2	0.28
Hind-limb paralysis	4	0.57
Emaciation	12	1.71
Gross health abnormalities	33	4.70

## Data Availability

The data has been supplied as a Appendix A along with the manuscript.

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
