# Peer review of "Stray Dogs and Public Health: Population Estimation in Punjab, India"

_vetsci, 2022, doi:10.3390/vetsci9020075_

Round 1

Reviewer 1 Report

Reviewer comments for manuscript ID vet sci -1568031 entitled ‘Stray dogs and public health: Population estimation in Punjab, India’

General Comments

An excellent and a very relevant study addressing a serious public health and animal welfare issue in India. The method of counting dogs adopted in this study is considered the most accurate assessment in public health studies. Statistical treatment of the data is thorough and unique. I am hopeful this study encourages further work on this very important public health concern in India on a wider scale. It will enable scientific approach in devising stray dog birth control programmes efficiently. The manuscript has been excellent written. I have very few corrections/suggestions to make.

Specific Comments

Page 2, para 4: Please insert ‘birth’ before control in ‘……..and a lack of consistently employed sustained control programs (Butcher, 1999)

Last para page 2: Refer ‘According to official data, there are 17.14 million stray dogs in India (BAHS, 2012). Please refer to a recent data on stray dog population in India’

Page 10, para 1: Can the authors please any other reason for this discrepancy in population figures of stray dogs?

Page 10, para 2: Can you please support this assertion with appropriate reason and reference –‘Many factors − such as higher male survivability or high mortality in female dogs − favour a high population of male dogs’

Author Response

Ref: vetsci-1568031
Title: Stray dogs and public health: Population estimation in Punjab, India
Journal: Veterinary Sciences

Dear Editor,

Thank you for considering our manuscript for publication in Veterinary Sciences. We have incorporated all the suggested changes into the manuscript.

 Our responses (non-italicised) can be found after reviewers’ original comment (in italics).

Kind Regards
Balbir B Singh

(Corresponding author)

Comments from the editors and reviewers:

Reviewer 1

General Comments

An excellent and a very relevant study addressing a serious public health and animal welfare issue in India. The method of counting dogs adopted in this study is considered the most accurate assessment in public health studies. Statistical treatment of the data is thorough and unique. I am hopeful this study encourages further work on this very important public health concern in India on a wider scale. It will enable scientific approach in devising stray dog birth control programmes efficiently. The manuscript has been excellent written. I have very few corrections/suggestions to make.

Authors’ response: Thank you for your comments. We have addressed all the points raised by you.

Specific Comments

Page 2, para 4: Please insert ‘birth’ before control in ‘……..and a lack of consistently employed sustained control programs (Butcher, 1999)

Authors’ response: Modified as suggested.

Last para page 2: Refer ‘According to official data, there are 17.14 million stray dogs in India (BAHS, 2012). Please refer to a recent data on stray dog population in India’

Authors’ response: The stray dogs count was estimated in the 19th Livestock census (BAHS, 2012). As far as we are aware, the objective information after that remains unavailable.

Page 10, para 1: Can the authors please any other reason for this discrepancy in population figures of stray dogs?

Authors’ response: Additional reasons for this discrepancy has also been provided (see below).

  As far as we are aware, this is the first systematic estimation of the stray dog population in Punjab. According to 2012 official data, there were estimated to be 305,482 stray dogs in Punjab (DAHP, 2012). However, we estimated the stray population to be 519,000 to 1569,000 using mark re-sight and the Superduplicates method, which is much higher than the official estimates. Mark re-sight is considered a practical way to accurately estimate the number and distribution of a stray dog population, if the assumption of a closed population is fulfilled (Seber, 1970; Cooch et al., 2014). In the present investigation, marking and subsequent counting events were completed within two days and thus the assumption of a closed population was likely valid because the period between the counting events was very short. Recent studies indicate that the SuperDuplicates method is also useful to estimate dog counts, particularly for two sample surveys conducted on consecutive days (Tiwari et al., 2018). The large difference between our estimate and the official estimate could be explained by the animal husbandry department having estimated the stray dog population size in 2011; there might have been a substantial increase in the stray dog population since then due to the lack of effective stray dog control programs. Differences in the methods used may also be responsible for difference in stray dog estimations. This is consistent with the findings of researchers in other areas (Davis and Menon, 2016).

Page 10, para 2: Can you please support this assertion with appropriate reason and reference –‘Many factors − such as higher male survivability or high mortality in female dogs − favour a high population of male dogs’

Authors’ response: The discussion has been elaborated on this issue (please see below).

Our estimate of a higher male to female dog population ratio is similar to the findings of other national and international studies (Totton et al., 2010; Tenzin et al., 2015 a; Pal, 2001; Mustiana et al., 2015). Whilst this might be due to methodology differences or measurement error, many factors that influence higher male survivability or higher mortality in female dogs favor a high population of male dogs. Populations with unequal sex ratio have a lower reproduction potential compared to equal sex ratio popu-lations (Shuster and Wade, 2003). The sex ratio of the owned dog population might be re-sponsible for this phenomenon. It has been reported that owners and farming communi-ties prefer male dogs due to guarding requirements, reduced nuisance behaviour during oestrus, and avoiding unwanted puppies when compared to adult females (Hossain et al., 2013; Margawani and Robertson 1995). However, this needs to be further investigated.  

Hossain, M., Ahmedb, K., Marmac, A.S.P., Hossaind, S., Alie, M.A., Shamsuzzamanf , A.K.M., Nishizonog, A., 2013. A survey of the dog population in rural Bangladesh. Preventive Veterinary Medicine 111, 134–138.

Margawani, K., Robertson, I., 1995. A survey of urban pet ownership in Bali. The Veterinary Record 137, 486–488, https://doi.org/10.1136/vr.137.19.486.

Reviewer 2

This study aimed to estimate the size of the stray dog population in Punjab, India using two methodologies: Lincoln-Petersen's with Chapman's correction and the SuperDuplicates method. Results showed some discordances as expected. I consider that the manuscript is well written, the introduction is interesting and well structured.

Authors’ response: Thank you for your comments. We have addressed all the points raised by you.

Please note that this sentence is not well interpreted, the quantity is out of context: "It has been estimated that a population of 100,000 dogs deposits between 3-11 tons of feaces daily in to the environment".

Authors’ response: Modified as suggested as follows:

The faecal shedding of pathogens by the stray dogs contaminates the environment (Beck, 1973), which is of substantial public and animal health concern.

I recommend a detailed review of the references throughout the text, I detected several errors that must be corrected before it is published if accepted:

Authors’ response: The Reference section has been thoroughly reviewed and inconsistencies in the references have been corrected.

Hogansen, 2013 is not well written in the text

Authors’ response: Corrected.

Webster is cited 2013 but in the list is 2010

Authors’ response: Corrected.

Butcher the reference is 2008 or 1999?

Authors’ response: Corrected.

Beck 1973, there is a recent version from 2002

Authors’ response: Corrected.

I could not find the Gibson reference in the list but is refered in the text, could not find the Neamtu, 1979 reference in the list

Authors’ response: The Gibson reference is in the reference section; the reference Neamtu, 1979 has been deleted.

Hossain et al is from 2013 or 2011?

Authors’ response: Corrected.

Tenzin et al 2005 they are two references please indicate along the text if is a or b, include them like that in the text.

Authors’ response: Modified as suggested.

Table 1 needs to be edited for better understanding.

Authors’ response: Modified as suggested (please see Tables 1 and 2).

I have some technical doubts. How long does the water-soluble spray that you used last? Is it possible that due to the rain this could be eliminated and that alter the values obtained in the study? Authors mentioned that the size of rural dogs was higher than the estimated.  Can the last of the mark influenced this?

Authors’ response: Before conducting this study, it was tested that the water-soluble spray lasts more than 72 hours. The study period was short (two days) for each specific village/ ward. Therefore, rainy days if any, were knowingly avoided.

With the Lincoln-Petersen’s index or the Chapman’s they should be sure that no owned dogs were recorded, since this is not a trustable way to determine if dogs are owned or not owned, maybe if they previously carried out a census of owned dogs the estimated size of the population could be more accurate. In fact, Tiwari et al 2018 do not recommend the Lincoln-Petersen’s index or the Chapman’s corrected estimator for FRD. Could you be so fine to justify why you use this method.

Authors’ response: Whilst the counts might include some owned dogs, we reduced this likelihood as follows (included in Methods, Page 4):

“The purpose of this study was to estimate the size of the stray dog population to inform ABC programs. Therefore, an announcement was made before the stray dog count in the relevant villages and wards about the stray dog counting to encourage people to leash their pet dogs, to avoid over-estimation of the population.”

We have also discussed this limitation (Discussion Page 11, final paragraph).

Yes, we agree with you that Lincoln-Petersen’s index or the Chapman’s corrected estimator is not the best method for FRD. Therefore, the dogs were additionally counted using SuperDuplicates method.

What is the contribution of this work in terms of decision-making, organization of campaigns for the control of diseases transmitted by dogs? 

Authors’ response: Please see our concluding paragraph. The estimated stray dog numbers can be used to guide development of a dog population control program in Punjab, especially if detailed information about resources (including costs) and logistics required can be included.

The authors mentioned that in 2012 the official data estimated 305, 482 stray dogs in Punjab, this is almost 10 years ago, but this study was performed between 2016-2017 How accurate are these results after five years?

Authors’ response: We agree with you that this study has been conducted between 2016-17. However, no studies or objective information on stray dog counts have been conducted in Punjab since 2016-2017, and the results will be valuable for dog population control program in the state especially if further data are collected, to determine the trend over time.

In the discussion authors explain that since 2011 the stray dog population size have substantially increased due the lack of effective stray dog control programs but then mentioned that the populations with unequal sex ratio (like in this study) have a lower reproduction potential, this is contradictory.

Authors’ response: Yes, we agree with you that unequal sex ratio (like in this study) have a lower reproduction potential, but this does not mean a decrease in dog population, although the rate of increase is likely to vary accordingly. It is possible that without this difference in sex ratio, the stray dog population in Punjab could have been much larger.

Reviewer 2 Report

This study aimed to estimate the size of the stray dog population in Punjab, India using two methodologies: Lincoln-Petersen's with Chapman's correction and the SuperDuplicates method. Results showed some discordances as expected.

I consider that the manuscript is well written, the introduction is interesting and well structured

Please note that this sentence is not well interpreted, the quantity is out of context: "It has been estimated that a population of 100,000 dogs deposits between 3-11 tons of feaces daily in to the enviroment".

The original references is "Feces usually dissapear within a week, but they can remain on the ground for over a month. The daily short term insult to the enviroment is inthe order of 2,700 to 10,000 kg (3 to 11 tons) of feces if we assume a population of 100,000 dogs"

I recommend a detailed review of the references throughout the text, I detected several errors that must be corrected before it is published if accepted:

Hogansen, 2013 is not well written in the text,
Webster is cited 2013 but in the list is 2010,
Butcher the reference is 2008 or 1999?,
Beck 1973, there is a recent version from 2002,
I could not find the Gibson reference in the list but is refered in the text, could not find the Neamtu, 1979 reference in the list
Hossain et al is from  2013 or 2011?,
Tenzin et al 2005 they are two references please indicate along the text if is a or b, include them like that in the text,

Table 1 needs to be edited for  better understanding

I have some technical doubts. How long does the water-soluble spray that you used last? Is it possible that due to the rain this could be eliminated and that alter the values obtained in the study? Authors mentioned that the size of rural dogs was higher than the estimated.  Can the last of the mark influenced this?

With the Lincoln-Petersen’s index or the Chapman’s they should be sure that no owned dogs were recorded, since this is not a trustable way to  determined if dogs are owned or not owned, maybe if they  previously carried out a census of owned dogs the estimated size of the population culd be more accurate. In fact, Tiwari et al 2018  do not recommend the Lincoln-Petersen’s index or the Chapman’s corrected estimator for FRD. Could you be so fine to justify why you use this method.

What is the contribution of this work in terms of decision-making, organization of campaigns for the control of diseases transmitted by dogs? 
The authors mentioned that in 2012 the official data estimated 305, 482 stray dogs in Punjab, this is almost 10 years ago, but this study was performed between 2016-2017 How accurate are these results after five years?

In the discussion authors explain that since 2011 the stray dog population size have substantially increased due the lack of effective stray dog control programs but then mentioned that the populations with unequal sex ratio (like in this study) have a lower reproduction potential, this is contradictory.

Author Response

(The authors gave the same response as above.)
